# Identity of neocortical layer 4 neurons is specified through correct positioning into the cortex

**Koji Oishi[1]\*[†], Nao Nakagawa[2,3], Kashiko Tachikawa[1], Shinji Sasaki[1], Michihiko Aramaki[1], Shinji Hirano[4], Nobuhiko Yamamoto[5], Yumiko Yoshimura[2,3], Kazunori Nakajima[1]\***

[1]Department of Anatomy, Keio University School of Medicine, Tokyo, Japan; [2]Division of Visual Information Processing, National Institute for Physiological Sciences, National Institutes for Natural Sciences, Okazaki, Japan; [3]Department of Physiological Sciences, Graduate University for Advanced Studies, Okazaki, Japan; [4]Department of Cell Biology, Kansai Medical University, Osaka, Japan; [5]Laboratory of Cellular and Molecular Neurobiology, Graduate School of Frontier Biosciences, Osaka University, Osaka, Japan

**Abstract** Many cell-intrinsic mechanisms have been shown to regulate neuronal subtype specification in the mammalian neocortex. However, how much cell environment is crucial for subtype determination still remained unclear. Here, we show that knockdown of *Protocadherin20* *(Pcdh20)*, which is expressed in post-migratory neurons of layer 4 (L4) lineage, caused the cells to localize in L2/3. The ectopically positioned "future L4 neurons" lost their L4 characteristics but acquired L2/3 characteristics. Knockdown of a cytoskeletal protein in the future L4 neurons, which caused random disruption of positioning, also showed that those accidentally located in L4 acquired the L4 characteristics. Moreover, restoration of positioning of the *Pcdh20*-knockdown neurons into L4 rescued the specification failure. We further suggest that the thalamocortical axons provide a positional cue to specify L4 identity. These results suggest that the L4 identity is not completely determined at the time of birth but ensured by the surrounding environment after appropriate positioning.

**\*For correspondence:** Koji.Oishi@crick.ac.uk (KO); kazunori@keio.jp (KN)

**Present address:** [†]The Francis Crick Institute Mill Hill Laboratory, London, United Kingdom

**Competing interests:** The authors declare that no competing interests exist.

## Introduction

The mammalian neocortex consists of six layers, each containing one or more distinct subtype of neurons that share cell morphology, birth dates and connections with other regions of the central nervous system (CNS) (*Greig et al., 2013*; *Leone et al., 2008*; *Molyneaux et al., 2007*). The neurons are born in the ventricular zone (VZ) and subventricular zone (SVZ), migrate radially toward the pial surface, to eventually come to reside beneath the marginal zone (MZ). Later-born neurons migrate past earlier-born neurons to more superficial layers, which results in the inside-out patterning of the cortical plate (CP) (*Angevine and Sidman, 1961*; *Berry and Rogers, 1965*; *Molyneaux et al., 2007*; *Rakic, 1974*).

The processes involved in the differentiation and specification of distinct neuronal subtypes during development of the neocortex have been under investigation for a long time. Several determinants of the identities of neuronal subtypes have been identified, including Tbr1 (*Hevner et al., 2001*), Fezf2 (*Chen et al., 2005a*; *2005b*; *Molyneaux et al., 2005*), Ctip2 (*Arlotta et al., 2005*), Satb2 (*Alcamo et al., 2008*; *Britanova et al., 2008*; *Leone et al., 2014*), Otx1 (*Weimann et al., 1999*), Brn1/2 (*Dominguez et al., 2013*; *McEvilly et al., 2002*; *Sugitani et al., 2002*), Sox5

**eLife digest** The outer surface of the brain (the neocortex) in mammals is formed out of neurons arranged into layers. These layers are laid down during embryonic development, and each layer has characteristic mix of distinct subtypes of neurons that have different forms and sizes.

Previous evidence suggests that neurons that are born at around the same time end up in the same layer of the cortex, and tend to have the same form. It is also possible that the environment a newborn neuron finds itself in influences its eventual form. However, many previous studies have investigated the function of molecules that work within the neurons, and the effect that a neuron's surroundings have on its development still remains largely unknown.

Neurons that contain a protein called protocadherin20 normally end up in layer 4 of the neocortex. Oishi et al. genetically engineered mouse embryos so that the production of protocadherin20 was reduced in these neurons whilst the neocortex formed. These neurons were also tagged with a fluorescent marker, so that their eventual position and shape in the brain could be tracked.

Examining the brains of the mice after they had been born showed that the tagged neurons ended up not in layer 4, but in layers 2 and 3 of the neocortex. What is more, these neurons now looked similar to other neurons in layer 2 and 3, as well as producing proteins and establishing connections consistent with their new location. However, further experiments that placed neurons with reduced levels of the protocadherin20 protein into layer 4 showed that these neurons acquire the characteristics of other layer 4 neurons, despite lacking a key layer 4 protein.

These results therefore suggest that the eventual form of a neuron is not determined just by when it is born, but also by the environment that it finds itself in. In future studies, it will be important to clarify the molecular mechanism that provides the appropriate environment and so regulates the identity of the neurons in the developing cortex.

(*Kwan et al., 2008*; *Lai et al., 2008*), and CoupTFI (*Tomassy et al., 2010*), which are transcription factors expressed typically in a neuronal subtype-specific manner. However, the mechanisms by which neurons obtain such subtype-specific characteristics during development still remain largely unclear.

Given that each layer of the neocortex is occupied by neurons born at around the same time (*Angevine and Sidman, 1961*; *Rakic, 1974*), the hypothesis of temporal regulation of the progenitor cells, according to which the subtype of neurons is specified depending on their birth date, has been widely accepted (*Dehay and Kennedy, 2007*; *Molyneaux et al., 2007*). Consistent with this hypothesis, we observed that the cortical neurons acquire a birth-date-dependent segregation mechanism before their somas reach the MZ (*Ajioka and Nakajima, 2005*). Previous transplantation studies of ferret cerebral cortical neurons suggest that the ultimate laminar fate (whether they eventually come to reside in the superficial layers or in the deep layers) is determined, at least to some extent, in the progenitor cells (*McConnell and Kaznowski, 1991*). Recent in vitro culture studies also suggested a cell-intrinsic mechanism of subtype specification of cortical neurons (*Eiraku et al., 2008*; *Gaspard et al., 2008*; *Shen et al., 2006*); however, only a limited number of subtype specific markers were applied in these studies, still leaving it an open question whether all laminar fates are intrinsically determined.

Although this contention had attracted little attention until recently, it is also conceivable that cortical lamination or appropriate cell positioning in the CP is required for full differentiation of neurons. The neocortex of mutant mice, such as homozygous mutant mice for $Reln^{rl}$ (also known as *reeler*), which contains an almost normal set of cortical neurons (*Dekimoto et al., 2010*; *Hevner et al., 2003*), but severely disorganized cortical lamination, shows some changes in subtype specification (*Polleux et al., 2001*). In addition, several subtype-specific characteristics of cortical neurons appear in the post-migratory phase, when the cortical layers begin to develop, implying that additional events of subtype specification may also take place. For example, pyramidal neurons begin to develop their apical dendrites only after the neurons settle beneath the MZ (*Bayer and Altman, 1991*; *Marin-Padilla, 1984*; *Tabata and Nakajima, 2001*). Furthermore, a number of subtype-

specific molecular markers can be detected only in post-migratory neurons (*Alcamo et al., 2008*; *Arlotta et al., 2005*; *Britanova et al., 2008*; *Kwan et al., 2008*; *Lai et al., 2008*). Importantly, there is a high degree of plasticity in the identity of postmitotic cortical neurons, which was revealed by ectopic expression of a subtype-specific transcription factor (*De la Rossa et al., 2013*). We previously reported a set of genes that are preferentially expressed in the superficial region of the developing CP, including the primitive cortical zone (*Sekine et al., 2011*), of the mouse, and proposed that several events critical for proper neuronal maturation and layer formation must take place beneath the MZ (*Sekine et al., 2012*; *Tachikawa et al., 2008*). However, it is still not established whether the neuronal subtype might also be affected by the microenvironment in the cortical layers after radial migration.

L4 of the neocortex is composed of several types of spiny neurons, including the spiny stellate cells (*Staiger et al., 2004*), that integrate thalamic inputs into cortical networks (*Lopez-Bendito and Molnar, 2003*; *Petersen, 2007*). In contrast, most of the other neocortical neurons transmit output signals to subcortical regions or other cortical regions such as the contralateral cortex (*Arlotta et al., 2005*; *Thomson and Bannister, 2003*). Thalamocortical axons terminate primarily in L4, and the signals received in L4 are transduced mainly to L2/3 neurons (*Lopez-Bendito and Molnar, 2003*; *Thomson and Bannister, 2003*). Reflecting their specific functions, L4 neurons show a round shape and granular morphology with multiple short dendrites, which is unique, and distinct from other neocortical "pyramidal" neurons. In addition to these unique morphological characteristics, several genes are also expressed preferentially in L4 neurons, such as *Rorb* (also known as *RORβ*), *Unc5d* and *Coup-tf1* (*Nakagawa and O'Leary, 2003*; *Pouchelon et al., 2014*; *Zhong et al., 2004*; *Zhou et al., 1999*), suggesting underlying some mechanisms that might regulate the specification of L4 neurons at a molecular level. In this study, we newly identified a non-clustered protocadherin, *Protocadherin20 (Pcdh20)*, which encodes a putative transmembrane protein, as a gene that begins to be expressed in immature postmitotic neurons, and is expressed in L4 neurons in the postnatal stages.

The protocadherin family of proteins, whose adhesion strength is usually very weak or undetectable, belong to the cadherin superfamily of calcium-dependent cell-cell adhesion molecules, and are widely expressed throughout the CNS (*Kim et al., 2007*; *Morishita and Yagi, 2007*; *Suzuki, 2000*; *Takeichi, 2007*). Although several members of this protein family, including Pcdh20, show layer-specific expression in the CP (*Kim et al., 2007*), it still remains largely unclear whether these genes may play a role in neocortical layer formation and subtype specification of neocortical neurons.

Here, we investigated the role of Pcdh20 during the development of L4. We found that Pcdh20 regulated the positioning of "future L4 neurons" into L4 via RhoA signaling, and that its knockdown not only caused malpositioning of these neurons into L2/3, but also caused them to acquire L2/3 characteristics. Recovery of positioning of the *Pcdh20*-knockdown neurons to L4 also rescued the defect of their specification. Furthermore, thalamocortical axons appeared to provide a positional cue to immature L4 neurons. Our data indicate that Pcdh20 plays an essential role in the specification of L4 neurons through regulating positioning of the cells after radial migration to beneath the MZ.

## Results

### Expression pattern of *Pcdh20* during cortical development

To identify the genes responsible for the post-migratory events during layer formation, we explored genes that are preferentially expressed in the superficial part of the CP, where the radial migration of neurons eventually ends, on embryonic day 16.5 (E16.5) and E18.5 (*Tachikawa et al., 2008*), and are expressed in a layer-specific manner in the neocortex on postnatal day 7 (P7). As a result, we found that *Pcdh20* is preferentially expressed in L4 on P7 (*Figure 1A–A''*). No layer-specific signals were detected with the sense probe in the P7 neocortex (*Figure 1A'''*). On E18.5, *Pcdh20* was expressed beneath the MZ (*Figure 1B,B'*) in the somatosensory cortex, where a large fraction of the "future L4 neurons" resides after radial neuronal migration (*Ajioka and Nakajima, 2005*) (see also *Figure 2H*). We only found weak expression of *Pcdh20* in the E14.0 and E16.5 neocortex (*Figure 1C,C',D,D'*), where "future L4 neurons" were being produced and were migrating (*Ajioka and Nakajima, 2005*). The expression levels of *Pcdh20* were also analyzed by quantitative

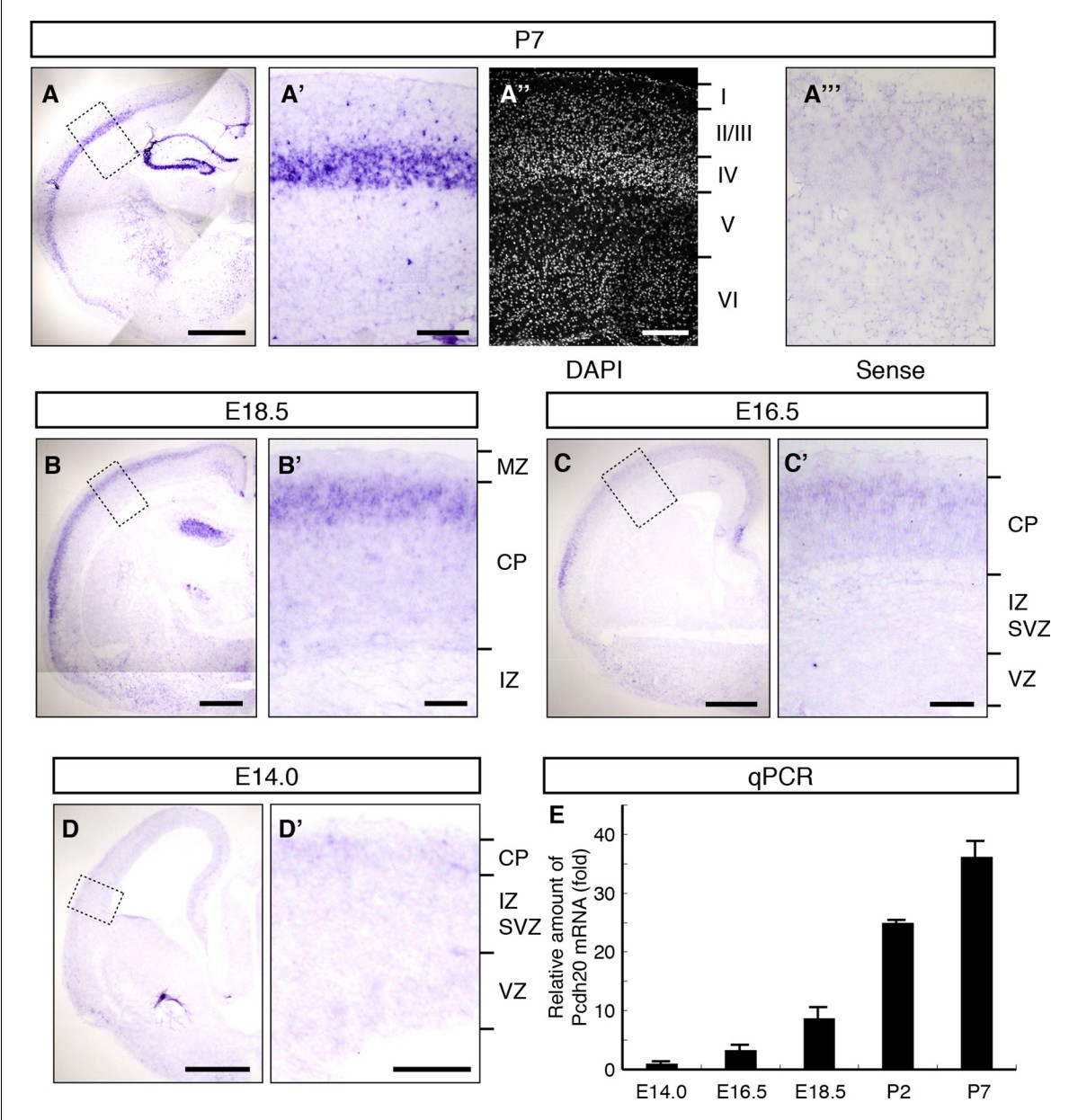

**Figure 1.** Expression of *Pcdh20* mRNA in the developing neocortex. (**A–D**) In situ hybridization for *Pcdh20* was performed in the E14.0, E16.5, E18.5 and P7 neocortex. The boxed regions in **A–D** are shown at higher magnification in **A'–D'**. Nuclear staining with DAPI of the section adjacent to **A'** shows the laminar structure of the neocortex (**A"**). No layer-specific signals were detected with the sense probe in the P7 neocortex (**A'''**). Expression of *Pcdh20* was weak in the E14.0 and E16.5 neocortex, but was clearly evident in the E18.5 neocortex; strong expression was observed in the P7 brain. (**E**) Quantitative RT-PCR analysis was performed at the indicated stages using *Pcdh20*-specific primers. Values are means ± SEM of three biological replicates. Scale bars, 1 mm (**A**); 500 µm (**B–D**); 200 µm (**A'**, **A"**); 100 µm (**B'–D'**). CP, cortical plate; IZ, intermediate zone; MZ, marginal zone; SVZ, subventricular zone; VZ, ventricular zone.

RT-PCR, and it was confirmed that the expression levels of *Pcdh20* mRNA in the early stages were much lower than those in the postnatal stages (***Figure 1E***). These results suggest that *Pcdh20* begins to be expressed strongly only at a relatively late stage of radial migration toward the MZ.

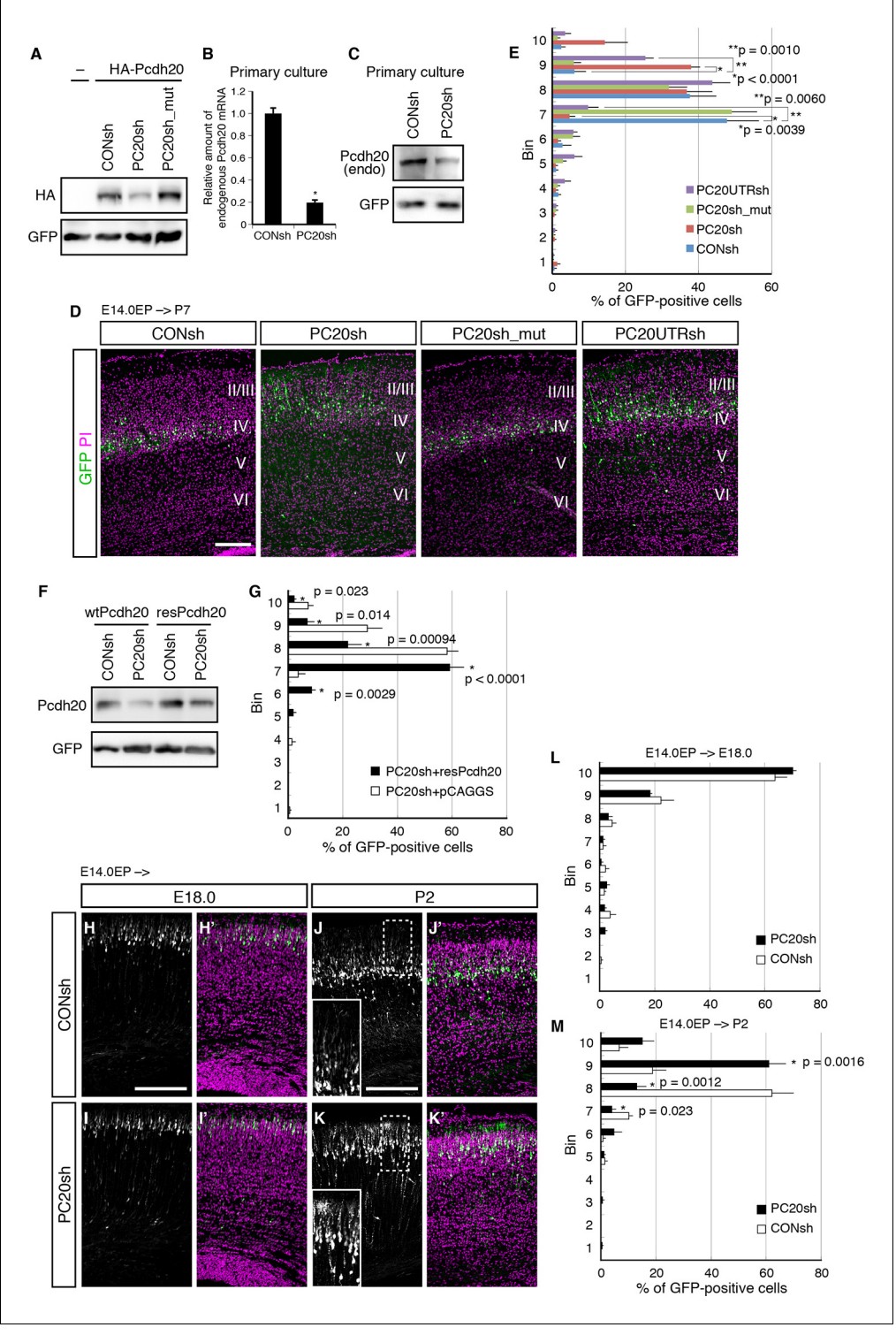

**Figure 2.** Pcdh20 is required for correct positioning of "future L4 neurons". (**A**) 293T cells were transfected with a control vector (CONsh), an shRNA vector targeting *Pcdh20* mRNA (PC20sh), or PC20sh_mut (which harbours point mutations in PC20sh) together with an HA-tagged Pcdh20 expression vector and a GFP expression vector. The cells were subjected to immunoblotting with antibodies to HA and GFP. (**B**) CONsh or PC20sh vector together with GFP vector was introduced on E14.0 cortices by in utero electroporation. Two days later, the cortices were removed, dissociated and cultured for 4 days in vitro. The GFP-positive cells were FACS sorted, and the amounts of *Pcdh20* mRNA were then analyzed by RT-qPCR. The *Pcdh20* levels were normalized by the expression of *β-actin*. Values are means ± SEM of three biological replicates. (**C**) CONsh or PC20sh vector together with GFP

*Figure 2 continued on next page*

*Figure 2 continued*

vector was introduced in dissociated E15.5 cortical cells by electroporation. Four days later, the cells were subjected to immunoblotting with antibodies to Pcdh20 and GFP. (**D, E**) CONsh, PC20sh, PC20sh_m, or PC20UTRsh vector together with a GFP vector was electroporated into the lateral ventricle of E14.0 embryos, then, the P7 brains were fixed and analyzed. The sections were counterstained with propidium iodide (PI, magenta). Most GFP-positive cells in the control experiment were located in L4, while the cells carrying the PC20sh or PC20UTRsh vector were located mainly in L2/3. Results of quantitative analyses of **D** are presented in **E** (n = 6 CONsh, n = 6 PC20sh, n = 5 PC20sh_mut, n = 5 PC20UTRsh). Details are described in Materials and Methods. (**F**) 293T cells were transfected with CONsh or PC20sh together with wild-type Pcdh20 (wtPcdh20) or a resistant form of Pcdh20 harboring mutations in the PC20sh-targeting site (resPcdh20). The cells were subjected to immunoblotting with antibodies to Pcdh20 and GFP. (**G**) PC20sh vector together with resPcdh20 was injected, and the brains were analyzed as in **E**. Results of quantitative analyses of *Figure 2—figure supplement 1A* are presented in **G** (n = 4 PC20sh+pCAGGS, n = 5 PC20sh+resPcdh20). (**H–K**) CONsh (**H, J**) or PC20sh (**I, K**) vector together with a GFP vector was electroporated into E14.0 brains, then, E18.0 (**H, I**) and P2 (**J, K**) brains were analyzed. The sections were counterstained with PI (magenta). The boxed regions are shown at higher magnification in the insets (**J, K**). Note that most GFP-positive cells with the PC20sh vector migrated normally, but were malpositioned in the P2 brains. (**L, M**) Quantitative data from E18.0 (**H, I**) (n = 4 CONsh, n = 4 PC20sh) and P2 (**J, K**) (n = 4 CONsh, n = 4 PC20sh) brains are presented. Scale bars, 200 μm (**D, H, J**).

The following figure supplement is available for figure 2:

**Figure supplement 1.** Pcdh20 is required for correct positioning of "future L4 neurons".

## Pcdh20 is required for correct laminar positioning of "future L4 neurons"

To investigate a possible role of Pcdh20 in the L4 formation, we utilized a vector-based RNA interference (RNAi) technique with short hairpin RNAs (shRNAs) to reduce the expression level of *Pcdh20* during cortical development. First, we examined the knockdown efficiency of the shRNA vectors on ectopically expressed Pcdh20. We found that expression of an shRNA vector targeting *Pcdh20* (hereinafter referred to as PC20sh) was associated with a markedly reduced protein expression level of Pcdh20 as compared with that of a control shRNA (CONsh) (*Figure 2A*). On the other hand, expression of a mutant shRNA vector harboring three point mutations in PC20sh (PC20sh_mut) did not significantly affect the expression level of Pcdh20 (*Figure 2A*). Furthermore, this knockdown vector was found to markedly decrease the endogenous expression levels of *Pcdh20* mRNA (*Figure 2B*) as well as protein (*Figure 2C*) in primary cortical cultures.

To examine the in vivo role of Pcdh20 during cortical development, we transferred RNAi vectors into living embryos by in utero electroporation (*Tabata and Nakajima, 2001*; *Tabata and Nakajima, 2003*). Various RNAi vectors together with a green fluorescence protein (GFP)-expressing vector were injected into the lateral ventricles of the mouse embryos on E14.0 and introduced into cortical cells by electroporation. First, the pups were sacrificed on P7, by which time, the basic structure of the neocortex was already expected to have formed. In the controls, most of the GFP-positive cells with CONsh or PC20sh_mut in the somatosensory cortex were located in L4 (*Figure 2D,E*). On the other hand, electroporation of PC20sh changed the laminar location of the GFP-positive cells to more superficial layers (*Figure 2D,E*). In addition, another shRNA vector targeting the 3'UTR of the *Pcdh20* gene also disrupted the laminar positioning of the electroporated cells (*Figure 2D,E*). The specificity of PC20sh for *Pcdh20* was further confirmed by an experiment in which co-introduction of an RNAi-resistant Pcdh20-expressing vector (resPcdh20) with PC20sh recovered the defect of neuronal positioning of the PC20sh-expressing cells (*Figure 2F,G*; *Figure 2—figure supplement 1A*). We also analyzed the effects of *Pcdh20* knockdown on deep layer neurons by transfecting the shRNA vectors on E12.5, when L5 and L6 neurons were expected to be produced. We found that *Pcdh20* knockdown in the deeper layer neurons hardly affected the cell positioning (*Figure 2—figure supplement 1B,C*), suggesting the specific function of Pcdh20 in L4 neurons. These results together suggest the requirement of Pcdh20 for correct positioning of the cells in L4.

This function of Pcdh20 appeared to be cell-autonomous, since sequential electroporation of mCherry fluorescent protein, followed by a mixture of GFP and PC20sh, showed that the *Pcdh20*

knockdown did not change the positioning of the mCherry-single positive control cells (*Figure 2—figure supplement 1D*).

Given that Pcdh20 is required for correct positioning of future L4 neurons, we next examined whether ectopic expression of Pcdh20 in other layers could change their positioning to L4. The results revealed that ectopic expression of Pcdh20 did not cause repositioning of L2/3 neurons to L4 (*Figure 2—figure supplement 1E,F*), but caused the same cells to be located more broadly in L2/3 as compared with the control cells (*Figure 2—figure supplement 1E,F*), suggesting that Pcdh20 expression by itself is not sufficient to change the location and morphology of L2/3 neurons to L4, and that some other factor(s) are also involved in the Pcdh20-dependent positioning of the L4 neurons.

## The effects of knockdown of *Pcdh20* became evident only after radial neuronal migration toward the MZ

The malpositioning of the neurons by *Pcdh20* knockdown could be caused by impaired neuronal migration, despite the very low expression levels of *Pcdh20* during the radial migration of the neurons. To determine whether the *Pcdh20* knockdown affected the neuronal positioning by inhibiting radial migration of the neurons toward the MZ or by a mechanism operative after the radial migration, a series of time-course experiments were performed. Brains electroporated on E14.0 were analyzed on E16.5, E18.0 and P2, the time-points roughly corresponding to mid-migration, end of migration and post-migration, respectively (*Figure 2H–K* and data not shown). On E16.5 and E18.0, few differences were observed between the control and *Pcdh20*-knockdown brains (*Figure 2H,I,L* and data not shown). In contrast, analysis of the P2 brains, in which the "future L4 neurons" had started to form a layer structure beneath the primitive L2/3, revealed that the *Pcdh20*-knockdown cells were located in a slightly more superficial position than that in the controls (*Figure 2J,K,M*). These data suggest that Pcdh20 may play an important role in the post-migratory stage, when the neurons are already present beneath the MZ and start to express Pcdh20 strongly.

## Acquisition of L2/3 characteristics of "future L4 neurons" induced by *Pcdh20* knockdown

We next characterized these ectopic "future L4 neurons" malpositioned in L2/3 using several subtype-specific molecular markers. First, we examined the expression of Rorb, a well-known L4 marker in the mature neocortex (*Nakagawa and O'Leary, 2003*). Immunohistochemical analysis of the P7 brains showed that while most of the GFP-positive cells in the control samples expressed Rorb, the percentage of Rorb-positive cells among the GFP-positive cells was markedly decreased by knockdown of *Pcdh20* (*Figure 3A–C*; *Figure 3—figure supplement 1A,B*). Expression of KCNH5 (a potassium voltage-gated channel, subfamily H), another marker of L4 neurons, was also not observed in the ectopically located GFP-positive cells by *Pcdh20* knockdown (*Figure 3D,E*; *Figure 3—figure supplement 1C,D*). Moreover, immunostaining for NetrinG1, a marker of thalamocortical axons (TCAs), showed that the malpositioned neurons by *Pcdh20* knockdown did not grossly change the growth of TCAs (*Figure 3—figure supplement 1K,L*). These results suggest that the malpositioned *Pcdh20*-knockdown neurons lost the characteristics of L4 neurons.

On the other hand, while the expression of Lhx2, a L2/3 marker (*Nakagawa et al., 1999*), was detected in only a small fraction of the GFP-positive neurons in the controls, *Pcdh20* knockdown was associated with a markedly increased percentage of Lhx2-expressing cells (*Figure 3F–H*; *Figure 3—figure supplement 1E,F*). We also found that *Pcdh20* knockdown was associated with an increase in the percentage of neurons positive for Satb2, a callosal neuron maker (*Alcamo et al., 2008*; *Britanova et al., 2008*) (*Figure 3I–K*; *Figure 3—figure supplement 1G,H*), and Tbr1, a L2/3 and L6 marker (*Hevner et al., 2001*) (*Figure 3L–N*; *Figure 3—figure supplement 1I,J*). These results suggest that *Pcdh20* knockdown caused the neurons destined for L4 to acquire L2/3 characteristics, at least at a molecular marker level.

We further investigated the stage at which misspecification of the laminar characteristics became evident during cortical development, by analyzing the time-course of changes in the expression of the molecular markers. We first analyzed the expression of Rorb and Lhx2 in the E18.0 neocortex. However, since Rorb expression was almost exclusively detected in L5, and Lhx2 expression was uniform in all the superficial layer neurons (both future L2/3 and L4) on E18.0 (*Figure 3—figure*

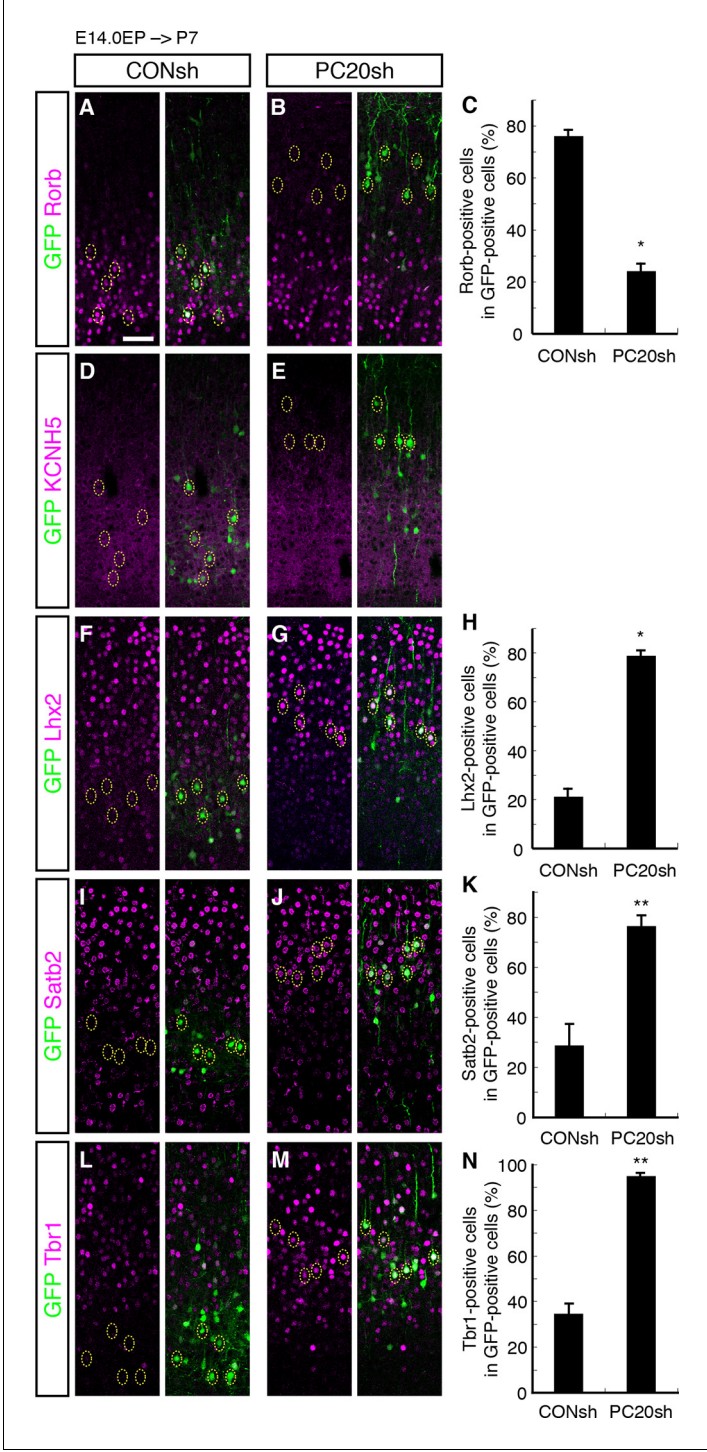

**Figure 3.** Acquisition of L2/3 molecular features of "future L4 neurons" induced by *Pcdh20* knockdown. (A–N) CONsh or PC20sh vector together with a GFP vector was electroporated into E14.0 brains, then, P7 brains were fixed and analyzed. The sections were immunostained for Rorb (A, B), KCNH5 (D, E), Lhx2 (F, G), Satb2 (I, J), and Tbr1 (L, M). The images at low magnification are shown in *Figure 3—figure supplement 1*. The results of quantitative analyses for Rorb (C), Lhx2 (H), Satb2 (K), and Tbr1 (N) are presented (n = 4 CONsh, n = 4 PC20sh). *Pcdh20*-knockdown cells showed diminished expressions of Rorb and KCNH5, but acquired expressions of Lhx2, Satb2 and Tbr1. *p < 0.0001, **p < 0.005. Scale bar, 50 μm (A).

The following figure supplements are available for figure 3:

*Figure 3 continued on next page*

*Figure 3 continued*

**Figure supplement 1.** *Pcdh20* knockdown resulted in "future L4 neurons" acquiring L2/3 characteristics.
**Figure supplement 2.** Postmitotic function of Pcdh20 regulates the development of L4 neurons.

---

*supplement 1M–P*), we could not assess the specification/differentiation of L2/3 and L4 at this stage. On P2, Rorb expression was detected in control L4 neurons, but not in *Pcdh20*-knockdown cells (*Figure 3—figure supplement 1Q,R*), similar to the finding in the P7 neocortex, although Lhx2 expression was still relatively uniform in all superficial layer neurons (*Figure 3—figure supplement 1S,T*), suggesting that at least some of the laminar characteristics had begun to be regulated by Pcdh20 by this stage.

We next analyzed whether the axonal projection patterns were affected by *Pcdh20* knockdown. Normally, many L4 neurons, which can be labeled by electroporation on E14.0, extend axons locally within the ipsilateral cortex, while most of the L2/3 neurons, which can be labeled by electroporation on E15.0, extend axons to the contralateral cortex (*Molyneaux et al., 2007*). We injected Fluoro-Gold, a fluorescent retrograde axonal tracer, into the hemisphere contralateral to the electroporated side to label commissural neurons, and analyzed whether the electroporated neurons incorporated this fluorescent dye. We first confirmed that the percentage of retrogradely labeled neurons in L2/3 was much higher than that in L4 (*Figure 4A,C,D*; *Figure 4—figure supplement 1A,C*). Interestingly, *Pcdh20* knockdown was associated with a substantial increase in the percentage of the retrogradely labeled neurons in the electroporated cells (*Figure 4B,D*; *Figure 4—figure supplement 1B*), suggesting that these axons projected to the contralateral cortex. These results suggest that the neurons labeled on E14.0, which normally populate L4, require Pcdh20 for proper L4 specification, and instead acquire L2/3 characteristics in the face of *Pcdh20* knockdown.

## Electrophysiological and morphological characterization of the *Pcdh20*-knockdown cells

We next compared the synaptic responses and membrane properties of the ectopic *Pcdh20*-knockdown neurons with those of the control L2/3 and L4 cells. Electrophysiological properties of control or *Pcdh20*-knockdown cells were examined in coronal slices of the somatosensory cortex prepared from the P14-19 brains that had been electroporated on E14.0 or E15.0. Spontaneous excitatory postsynaptic currents (sEPSCs) were clearly detected in the *Pcdh20*-knockdown cells (*Figure 4H*; *Figure 4—figure supplement 1D–F*), indicating that the malpositioned cells also received synaptic inputs. We also found that the *Pcdh20*-knockdown cells differed in some, if not in all, characteristics from the control L4 cells, while being similar in some characteristics to the L2/3 cells (*Figure 4H*). These characteristics that conferred resemblance to the L2/3 neurons included the frequency of sEPSCs, half width of the action potential, and the input resistance (*Figure 4H*). These data suggest that the electrophysiological properties of the malpositioned cells by *Pcdh20* knockdown came to resemble those of the surrounding control L2/3 cells.

Histochemical visualization with biocytin loaded through the recording patch pipette revealed that almost all control L4 neurons showed the morphology of stellate neurons, which extend their dendrites radially in many directions (*Figure 4E*; *Figure 4—figure supplement 2A*). On the other hand, the malpositioned *Pcdh20*-knockdown neurons possessed a thick apical dendrite with terminal arborization in the L1 and a thin axon extending toward the ventricular side (*Figure 4F*; *Figure 4—figure supplement 2B*), similar to the control pyramidal neurons in L2/3 (*Figure 4G*; *Figure 4—figure supplement 2C*). These morphological changes were also observed in the magnified images of GFP fluorescence obtained on P7 (*Figure 4—figure supplement 2D–F*).

These changes in the electrophysiological properties and cell morphology, together with those in the expressions of molecular markers, strongly suggest that knockdown of *Pcdh20* expression cause the neurons destined for L4 to acquire L2/3 neuronal characteristics. However, this conversion was incomplete, because the membrane capacitance of the *Pcdh20*-knockdown cells was almost the same as that of the control L4 cells and less than a half of that of the control L2/3 cells (*Figure 4H*); thus, the *Pcdh20*-knockdown cells still remained small in size, like the L4 neurons. We also noticed

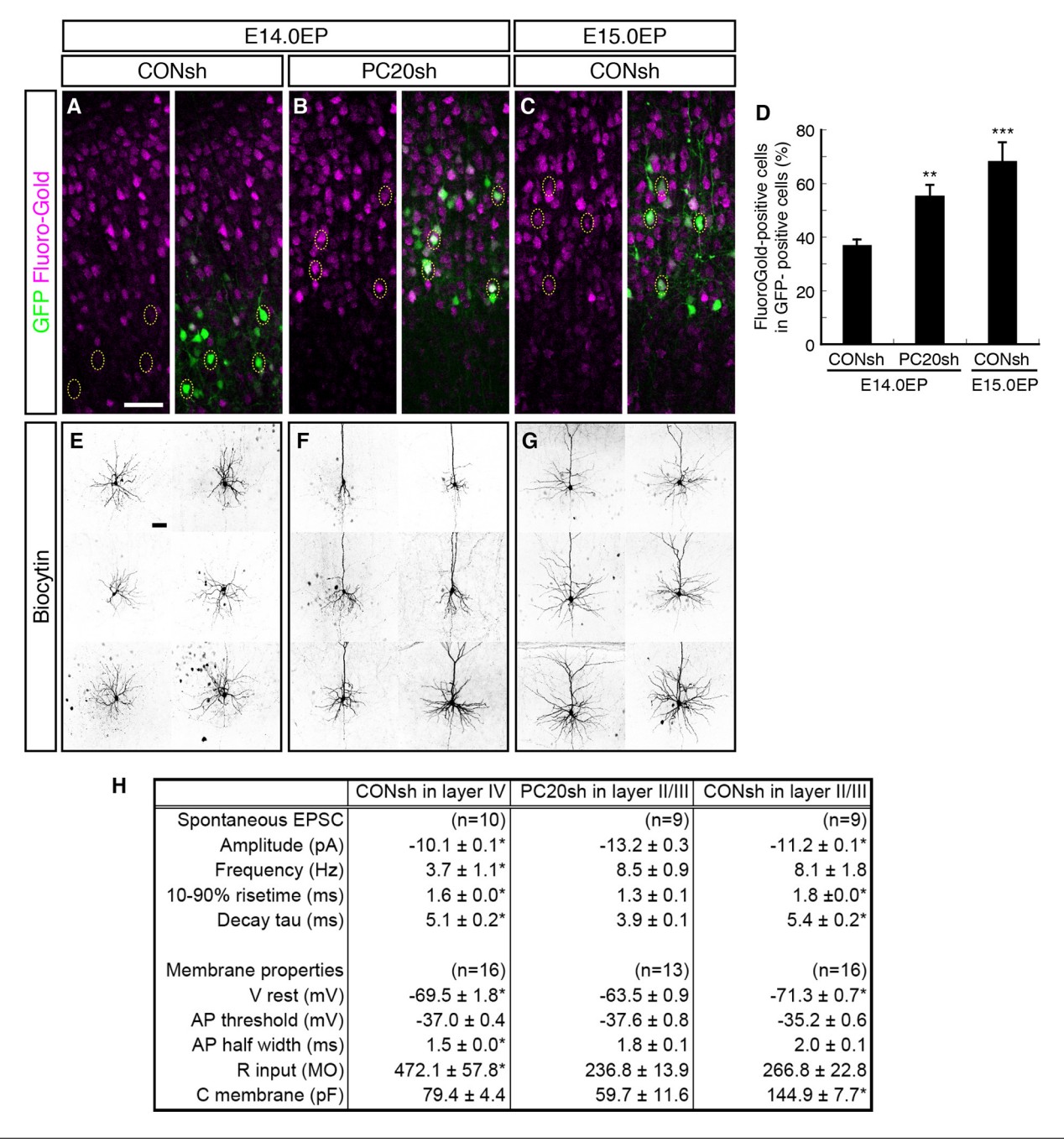

**Figure 4.** Alterations of axon projections and electrophysiological properties by *Pcdh20* knockdown. (**A–D**) CONsh (**A, C**) or PC20sh (**B**) vector together with a GFP vector was electroporated into E14.0 (**A, B**) or E15.0 (**C**) brains. Fluoro-Gold was injected at P7 into the cortex contralateral to the side of electroporation, then, the P10 brains were fixed and analyzed. The images at low magnification are shown in *Figure 4—figure supplement 1*. Results of quantitative analyses of **A–C** are presented in **D** (n = 4 each group). (**E–G**) CONsh (**E, G**) or PC20sh (**F**) vector together with a GFP vector was electroporated into E14.0 (**E, F**) or E15.0 (**G**) brains. The cell morphologies were then analyzed in P14–19 brains by injection of biocytin and sequential visualization with avidin-Cy3. The remaining cells are presented in *Figure 4—figure supplement 2A–C*. *Pcdh20*-knockdown neurons possessed thick apical dendrites, a characteristic morphologic feature of L2/3 neurons, but not of L4 neurons. (**H**) Effects on electrophysiological properties by *Pcdh20* knockdown. Experiments were performed as described in **E–G**, and the electrophysiological properties were analyzed at P14-19. *p < 0.05 (compared with the *Pcdh20*-knockdown brains electroporated on E14.0), **p < 0.005, ***p = 0.0204 (compared with the control brains electroporated on E14.0). Scale bars, 50 μm (**A, E**).

The following figure supplements are available for figure 4:

*Figure 4 continued*

**Figure supplement 1.** sEPSCs in *Pcdh20*-knockdown cells and control L2/3 and L4 cells.
**Figure supplement 2.** Morphological changes of "future L4 neurons" induced by knockdown of *Pcdh20*.

that *Pcdh20*-knockdown neurons tended to possess less well-developed basal dendrites as compared to the control L2/3 neurons (*Figure 4F,G*; *Figure 4—figure supplement 2*).

## Postmitotic functions of Pcdh20 regulate L4 specification

The timing of terminal mitosis has been supposed to be critical for determination of the ultimate laminar fates (*Dehay and Kennedy, 2007*; *Molyneaux et al., 2007*). It is thus possible that misspecification of the neuronal laminar identity after *Pcdh20* knockdown was caused as a result of additional or delayed cell divisions. To address this question and determine whether the laminar specification was indeed changed by *Pcdh20* knockdown in the postmitotic cells, we performed electroporation of the shRNA and GFP vectors on E14.0 and subsequently labeled the entire population of mitotic cells by serial injections of BrdU every 5 hr for 20 hr immediately after the electroporation (*Figure 3—figure supplement 2A*). In this experiment, the population of GFP-positive and BrdU-negative cells (referred to as GFP$^+$/BrdU$^-$ cells) was thought to represent postmitotic neurons or cells in the very late stages of the cell cycle in terminal mitosis in the VZ at the time of the electroporation, based on the following reasons. First, plasmid vectors are incorporated primarily into VZ cells by in utero electroporation. Second, since the duration of the S phase at this stage is about 4 hr, and since the nuclei in the S phase can be labeled for at least 2 hr after a single injection of BrdU (*Takahashi et al., 1992*), injections of BrdU at intervals of up to 6 hr are sufficient for labeling all the mitotic VZ cells. Finally, the total length of the cell cycle is about 15 hr, which is shorter than the total period of BrdU application (see ref. (*Tabata et al., 2009*) for more details). These GFP$^+$/BrdU$^-$ cells are included in the "slowly exiting population", a population that becomes postmitotic in the VZ and migrates out slowly into the multipolar cell accumulation zone above the VZ (*Tabata et al., 2009*).

Triple immunostaining of P7 brain sections for GFP, BrdU and Rorb revealed that the percentage of Rorb-positive cells in the GFP$^+$/BrdU$^-$ population was markedly decreased by knockdown of *Pcdh20* (*Figure 3—figure supplement 2B,C*), suggesting that *Pcdh20* knockdown perturbed L4 specification of the postmitotic cells. The extent of decrease in the percentage of Rorb-positive cells was comparable in all cell populations (*Figure 3C*), suggesting that the effect of *Pcdh20* knockdown was not influenced by whether the knockdown vector was introduced into a postmitotic population or a mitotic population. Importantly, the proportions of BrdU-negative cells in the entire population of GFP-positive cells were almost the same between the control and knockdown cells (*Figure 3—figure supplement 2D*), suggesting that *Pcdh20* knockdown did not alter cell cycle progression in the VZ/SVZ cells.

We further investigated whether the postmitotic expression of Pcdh20 may be essential for its functions, by conducting a rescue experiment using a neuron-specific Tα1 (α1-tubulin) promoter-driven expression vector to express resPcdh20 (*Gloster et al., 1994*). We found that a Tα1-driven resPcdh20 expression vector rescued the phenotype of malpositioning of the cells caused by *Pcdh20* knockdown (*Figure 3—figure supplement 2E,F*), further suggesting the important function of Pcdh20 in postmitotic neurons.

## Involvement of RhoA signaling in the Pcdh20-dependent positioning of L4 neurons

We next sought to elucidate the molecular mechanisms involved downstream of Pcdh20. Given that other protocadherins have been reported to regulate small GTPase RhoA (*Chung et al., 2007*; *Unterseher et al., 2004*) and that Pcdh20 was demonstrated to exert an initial effect on neuronal morphogenesis, including dendritic morphology (*Figure 2J,K*), which could be regulated by RhoA (*Redmond and Ghosh, 2001*), we focused on RhoA signaling.

We first examined whether RhoA signaling was required for correct cell positioning of the L4 neurons. We inhibited RhoA signaling in the future L4 neurons by introducing a dominant-negative form of RhoA (T19N, DN-RhoA). The DN-RhoA-electroporated neurons were located more superficially as

compared with the control neurons (*Figure 5A,B*), similar to the effect of *Pcdh20* knockdown, suggesting that these two genes may function in the same signaling pathway to ensure correct positioning of L4 neurons.

Modification of RhoA signaling might affect radial neuronal migration toward the brain surface, although its role in radial migration is controversial; inhibition of RhoA interfered with neuronal migration in some studies (*Pacary et al., 2011*; *Xu et al., 2015*), but it promoted migration in other cases (*Cappello et al., 2012*; *Ge et al., 2006*; *Nguyen et al., 2006*). These previous discrepancies might have been caused by the differences in their experimental conditions. We examined the effect of DN-RhoA at E18.0, when future L4 neurons just finished migration and started to mature. We found that there was no significant difference in cell positioning between control and DN-RhoA expressing cells at least at the concentration/condition we used (*Figure 5—figure supplement 1*), suggesting that the DN-RhoA-expressing neurons settle in the ridge of the CP similarly to normal cells after radial migration, although they could have migrated differently. Therefore, the phenotype observed at P7 (*Figure 5A*) was thought to be caused mainly by the event after radial migration toward the brain surface is completed. However, we do not exclude the possibility that there might be an additional effect of migration difference on subtype specification, especially in the case of enhanced migration, because there was a tendency (not statistically significant) for the DN-RhoA-expressing cells to position in deeper layers at P7 (*Figure 5A,B* (bins 4–6)). One might suspect that accelerated neuronal migration could cause premature differentiation, which would result in more cells in deeper layers, although this effect of DN-RhoA, if any, was thought to be minor.

If RhoA is an intracellular effector downstream of Pcdh20, the phenotypes observed by *Pcdh20* knockdown should be rescued by activation of RhoA. To test this, we examined whether a constitutively active form of RhoA (CA-RhoA) could restore the phenotypes resulting from *Pcdh20* knockdown. The results revealed that co-expression of CA-RhoA with PC20sh indeed rescued the malpositioning of future L4 neurons caused by *Pcdh20* knockdown (*Figure 5C,D*). Moreover, we examined the misspecification phenotype associated with *Pcdh20* knockdown and found that the loss of Rorb expression by *Pcdh20* knockdown was also restored by co-expression of CA-RhoA with PC20sh (*Figure 5E–H*). These results suggest that RhoA functions downstream of Pcdh20.

## Pcdh20 specifies L4 identity through controlling cell positioning

The phenotypic consequences of *Pcdh20* knockdown, disrupted neuronal positioning and change of layer identity, raise at least three possibilities. One is that the two phenotypes might arise independently, and the other two are that the disrupted neuronal positioning may affect the neuronal specification, or vice versa. To distinguish among these possibilities, we sought to disrupt the neuronal positioning of "future L4 neurons" using another method. We took advantage of knockdown of *Doublecortin (Dcx)*, since knockdown of this gene in the mouse neocortex has been reported to cause random disruption of neuronal positioning (*Baek et al., 2014*; *Ramos et al., 2006*). We focused on the cells that came to reside in the superficial layers (L2–4) for the purpose of this study. The proportions of Rorb- as well as KCNH5-positive cells among the GFP-positive cells located in L2–4 were dramatically decreased by *Dcx* knockdown (*Figure 6A–E*; *Figure 6—figure supplement 1A–D*). Correspondingly, the proportions of Lhx2- as well as Satb2-positive cells were markedly increased by *Dcx* knockdown (*Figure 6F–K*; *Figure 6—figure supplement 1E–H*), implying the importance of cell positioning. On the other hand, no change in the proportion of Tbr1-positive cells was noted in association with *Dcx* knockdown (*Figure 6L–N*; *Figure 6—figure supplement 1I,J*), suggesting that Tbr1 expression may not be regulated simply by cell positioning. To investigate the relationship between subtype specification and the ultimate laminar positioning, we further examined the proportions of Rorb-positive cells among *Dcx*-knockdown neurons residing in L2/3 or L4. The majority of the neurons positioned in L4 expressed Rorb (72%), whereas none of the neurons in L2/3 expressed this marker (*Figure 6B*). These results suggest that the ultimate cell positioning determines most of the neuronal characteristics of the neurons labeled on E14.0, that is, into L2/3 or L4 neurons.

We also examined whether the neuronal malpositioning might be the major mechanism underlying the misspecification of the laminar identity of the *Pcdh20*-knockdown neurons. If neuronal malpositioning itself is critical for the misspecification of the *Pcdh20*-knockdown neurons, then this misspecification should be rescuable by recovering positioning of the knockdown neurons into L4. We therefore electroporated both *Pcdh20*- and *Dcx*-knockdown vectors to induce random localization of the electroporated neurons in the CP and analyzed the subtypes of the neurons accidentally

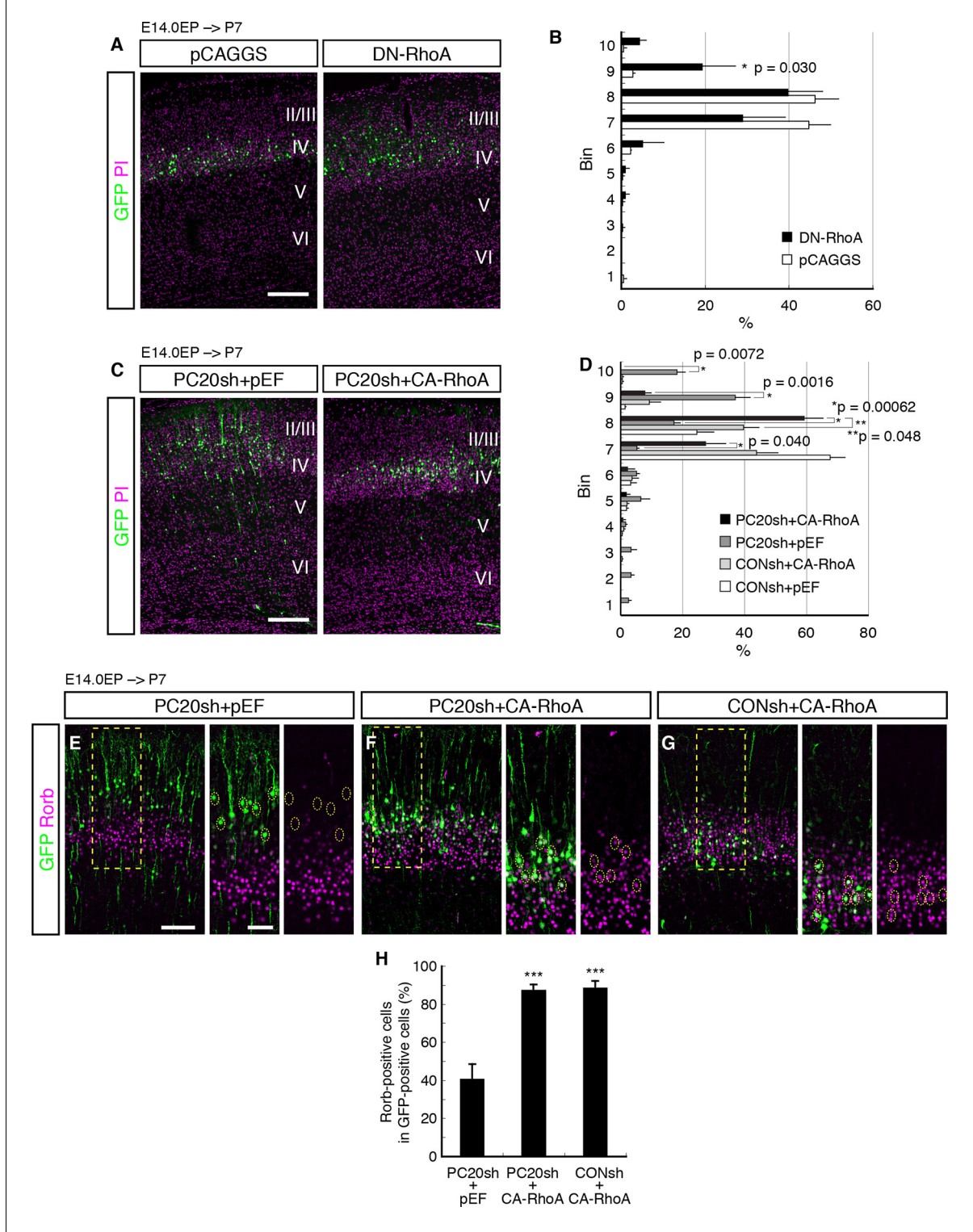

**Figure 5.** Involvement of RhoA in the Pcdh20-dependent development of L4 neurons. (**A, B**) The empty vector (pCAGGS) or dominant-negative RhoA expression vector (DN-RhoA) was electroporated into E14.0 brains, and P7 brains were analyzed. Results of quantitative analyses of **A** are presented in **B** (n = 4 pCAGGS, n = 4 DN-RhoA). Many of the DN-RhoA-expressing cells were located in L2/3. (**C, D**) The indicated vectors were electroporated into E14.0 brains, and P7 brains were analyzed. Results of quantitative analyses of c are presented in **D** (n = 4 each group). Constitutively active RhoA (CA-RhoA) rescued the malpositioning caused by the knockdown of *Pcdh20*. (**E–H**) The sections prepared from the brains in **C** were immunostained for Rorb and GFP. Results of quantitative analyses of **E–G** are presented in **H** (n = 4 PC20sh+pEF, n = 4 PC20sh+CA-RhoA, n = 5 CONsh+CA-RhoA). Active

*Figure 5 continued on next page*

*Figure 5 continued*

RhoA restored the specification failure induced by *Pcdh20* knockdown. ***p < 0.005. Scale bars, 200 µm (**A, C**); 100 µm (**E** on the left); 50 µm (**E** on the right).

The following figure supplement is available for figure 5:

**Figure supplement 1.** DN-RhoA did not change the cell positioning in early stages.

positioned in L4 in the absence of *Pcdh20*. We found that the neurons were indeed positioned in a random manner throughout the neocortex (*Figure 6—figure supplement 1N–P*), and that many of the neurons that were accidentally positioned in L4 expressed Rorb even in the absence of *Pcdh20* (*Figure 6O–Q*), suggesting that Pcdh20 controls L4 specification indirectly.

To further test if Pcdh20 has a direct effect on cell fate or not, we examined subtype specification in an in vitro primary cortical culture, in which effects from the in vivo cell environment are thought to be excluded. We cultured future L4 neurons from E18.0 cortices, at which time almost no obvious changes associated with *Pcdh20* knockdown are observed (*Figure 2H,I*; *Figure 3—figure supplement 1M–P*), and looked at the subtype specification after 4 days in vitro, when these cells become Rorb-positive in vivo. We found that *Pcdh20* knockdown did not change the percentage of Rorb-positive cells (*Figure 6R*), suggesting that Pcdh20 requires the in vivo cell environment to exert its function on the L4 specification. These results altogether strongly suggest that Pcdh20 specifies L4 identity through controlling the cell positioning of future L4 neurons. In addition, in a similar culture condition, *Pcdh20*-knockdown neurons were found to possess a higher number of neurites than the control neurons (*Figure 6S–U*), suggesting that Pcdh20 might, at least to some extent, directly regulate the neuronal morphology.

## TCAs play an important role in the subtype specification of L4 neurons

We finally explored the mechanisms underlying the establishment of L4 specification through cell positioning. We focused on TCAs, because the timing of invasion of these axons into the neocortex and the immature L4 (perinatal stages) is well correlated with that of maturation of the L4 neurons (*Lopez-Bendito and Molnar, 2003*). It is thus possible that TCAs invading upward from the deeper part of the CP may provide a positional cue to the cortical neurons. We therefore analyzed *Pcdh10* (also known as *OL-protocadherin* or *OL-pc*)-knockout mice, in which the TCAs are severely disturbed (*Uemura et al., 2007*). First, we examined the extent to which the TCAs are lost in the neocortex of this mutant line. Immunohistochemistry for NetrinG1, a TCA-specific marker, revealed a marked decrease of the TCAs in the brains of the *Pcdh10 (OL-pc)* knockout mice (*Figure 7A,B*). We next examined the expressions of several subtype-specific markers to investigate layer identities on P3 and P7. We found that expression of Rorb was markedly decreased in *Pcdh10 (OL-pc)* knockout mice (*Figure 7C–E*). In parallel with the decrease in the expression of the L4 marker, expressions of Lhx2 and Tbr1, both of which are not expressed in L4 in the control neocortex, were increased in the mutant neocortex (*Figure 7F–K*). These results suggest that the neurons that are destined to become L4 neurons lose their L4 identity and instead acquire some of the L2/3 characteristics, presumably because of the absence of the TCAs. Moreover, these results are similar to the misspecification phenotype of the malpositioned neurons by *Pcdh20* or *Dcx* knockdown, suggesting that TCAs might provide a positional cue to immature L4 neurons.

## Discussion

This study provides the first evidence to show that the eventual positioning of the neurons in the neocortex contributes to the acquisition of the layer-specific characteristics of the cells. We propose that the laminar identities of neurons within the superficial layers (L2–4) of the CP are not completely specified until the neurons arrive beneath the MZ; the ultimate fates of the L2/3 and L4 neurons are specified in the post-migratory phase according to the positioning of the cells. In this model, "future L4 neurons" are positioned in the deeper part of the superficial layers in a Pcdh20-RhoA-dependent manner after radial migration to the top of the CP, and then differentiate into mature L4 neurons, presumably by receiving signals from TCAs (*Figure 7L*). Although post-migratory neurons were

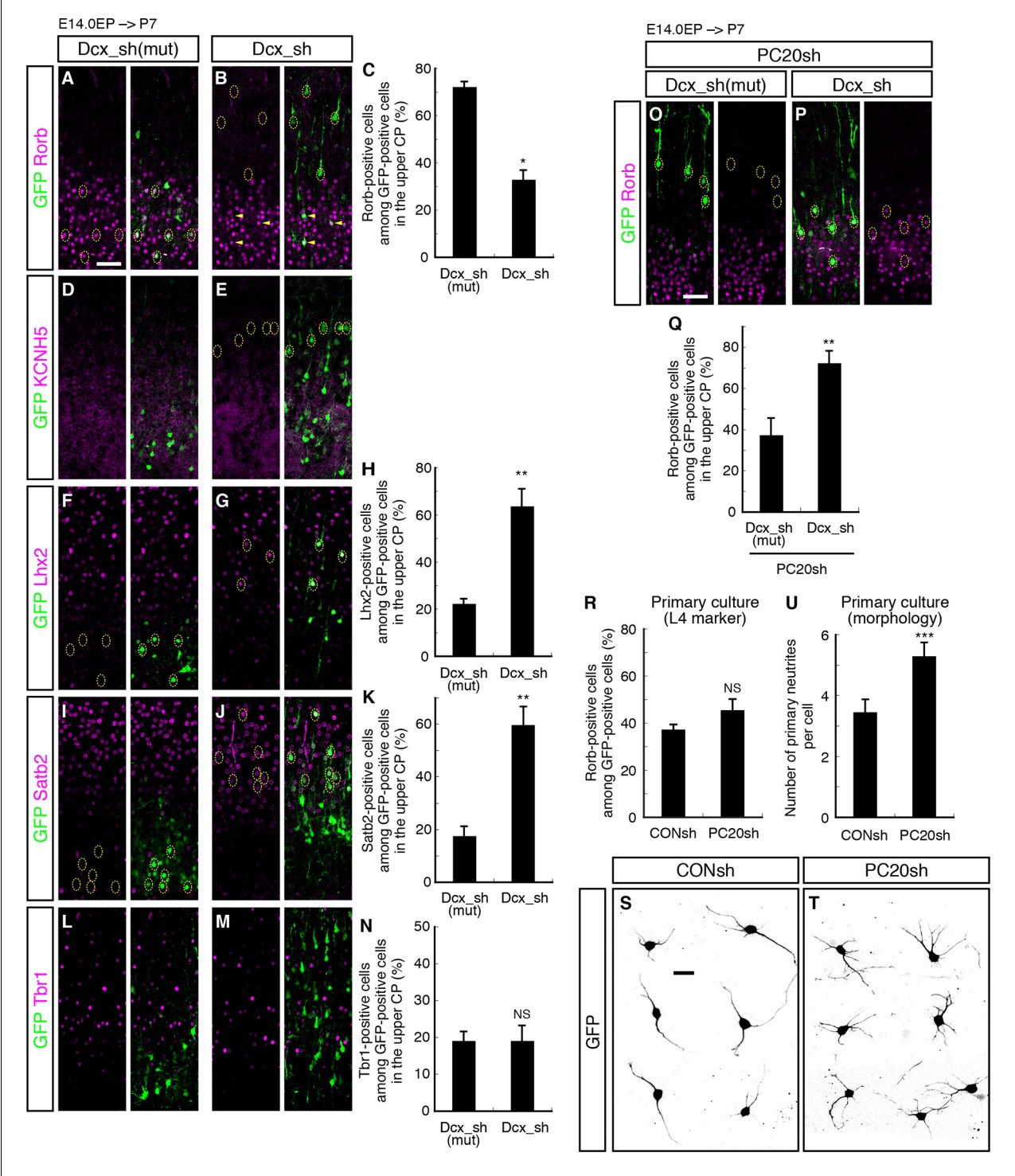

**Figure 6.** Failure of subtype specification of the cells positioned ectopically in L2/3 by knockdown of *Dcx*. (A–N) Control (Dcx_sh(mut), harboring point mutations in Dcx_sh) or Dcx_sh vector together with a GFP vector was electroporated into E14.0 brains, then, P7 brains were fixed and analyzed. The sections were immunostained for Rorb (A, B), KCNH5 (D, E), Lhx2 (F, G), Satb2 (I, J) and Tbr1 (L, M). The images at low magnification are shown in *Figure 6—figure supplement 1*. The results of quantitative analyses for Rorb (C), Lhx2 (H), Satb2 (K) and Tbr1 (N) are presented (n = 4 Dcx_sh(mut), n = 4 Dcx_sh). *Dcx*-knockdown cells exhibited some of the phenotypes observed in the *Pcdh20* knockdown experiment. (O–Q) Dcx_sh(mut) or Dcx_sh vector together with the PC20sh vector was electroporated into E14.0 brains, then, P7 brains were fixed and analyzed. The sections were immunostained for Rorb and GFP (O, P). The results of quantitative analyses are presented in Q (n = 4 Dcx_sh(mut)+PC20sh, n = 4 Dcx_sh+PC20sh). The cells showing recovery of positioning in L4 following introduction of the PC20sh and Dcx_sh vectors also showed recovery of Rorb expression. (R–

*Figure 6 continued on next page*

*Figure 6 continued*

U) CONsh or PC20sh vector together with a GFP vector was electroporated into E14.0 brains, then, E18.0 brains were dissociated. The neurons were cultured for 4 days (R) or 2 days (S–U) and immunostained for GFP and Rorb (R) or GFP (S–U). Results of quantitative analyses are presented in R (n =3 CONsh, n =3 PC20sh) and U (n = 22 CONsh, n = 21 PC20sh). *p < 0.0001, **p < 0.02, ***p < 0.005. Scale bars, 50 µm (A, O); 20 µm (S).
The following figure supplement is available for figure 6:

**Figure supplement 1.** Failure of subtype specification of the cells positioned ectopically in L2/3 by knockdown of *Dcx*.

generally thought to just pile up on the earlier-settled neurons after radial migration, our results imply that the processes that take place after radial migration are also precisely regulated and are critical for correct laminar positioning of at least the "future L4 neurons".

Sufficient attention has not been paid in previous studies to the relationship between cell positioning and laminar identity specification (*Dehay and Kennedy, 2007*; *Ramos et al., 2006*). In previous studies, the *Reln^{rl}/Reln^{rl}* neocortex, in which the cortical laminar structure is largely inverted and considerably disorganized, was shown to contain an almost normal set of neuronal subtypes (*Dehay and Kennedy, 2007*; *Hevner et al., 2003*; *Polleux et al., 1998*). Another relevant observation is that when future L2/3 neurons were induced to localize in L5, they did not acquire the characteristics of L5 neurons (*Ramos et al., 2006*). Here, we showed regulation by Pcdh20 of the cell positioning of "future L4 neurons" and also specification of their laminar identity, and concluded the causal effect of cell positioning regulated by Pcdh20 on L4 specification, which was supported by the results of the following experiments. First, malpositioning of the neurons by another method (i. e., knockdown of *Dcx*) reproduced the failure of the laminar identity specification of the L4 neurons. Second, loss of Pcdh20 caused misspecification of L4 neurons only in vivo but not in an in vitro primary culture, where positional effects did not exist. Third, recovery of cell positioning into L4 even in the absence of *Pcdh20* rescued the L4 characteristics. One might suspect that this model is controversial to the observation in the *Reln^{rl}/Reln^{rl}* cortex, where L4 neurons are actually generated. In this mutant, a substantial amount of TCAs was reported to grow into the mutant CP, although they pass a bizarre course (*Caviness and Frost, 1983*), which could be enough to induce L4 specification.

Although it has long been supposed that cell fate decision of cortical neurons takes place mainly in the progenitor cells (*Dehay and Kennedy, 2007*; *Molyneaux et al., 2007*), our data now suggest that L4 specification may also occur in postmitotic neurons, which is supported by several lines of evidence. First, the expression of *Pcdh20* mRNA was undetectable or very low, if any, in progenitor cells, but became clearer in the postmitotic neurons beneath the MZ, as evaluated by in situ hybridization and higher by quantitative RT-PCR. Consistent with this finding, the distribution of the *Pcdh20*-knockdown neurons remained normal before and during radial neuronal migration, but became disrupted after the neurons settled beneath the MZ. Moreover, introduction of the *Pcdh20*-knockdown vector even in a postmitotic population caused misspecification of the L4 neurons. Finally, expression of Pcdh20 in the postmitotic neurons was able to rescue the phenotypes induced by *Pcdh20* knockdown. This idea, of postmitotic regulation of subtype specification, is also supported by the fact that many subtype-specific genes play roles in postmitotic neurons (*Alcamo et al., 2008*; *Arlotta et al., 2005*; *Britanova et al., 2008*; *Kwan et al., 2008*; *Lai et al., 2008*; *Leone et al., 2014*) and that there is a high level of plasticity in the identity of postmitotic cortical neurons (*De la Rossa et al., 2013*).

Surprisingly, as the result of the misspecification induced by *Pcdh20* knockdown, which caused malpositioning of the "future L4 neurons" into L2/3 and loss of both the characteristic morphology of L4 neurons and of the expression of the L4 markers, the *Pcdh20*-knockdown neurons exhibited the morphological characteristics of pyramidal neurons and the functional characteristics of L2/3 neurons, including the characteristic marker expressions and axonal projections. These observations suggest that "future L4 neurons" have the capacity to differentiate into L2/3 neurons even in the postmitotic stage. Our results, however, also suggest regulation, at least in part, of the laminar fates in the progenitor cells, because the *Pcdh20*-knockdown "future L4 neurons" had "imperfect" characteristics as compared to the normal L2/3 neurons, such as poorly developed basal dendrites and relatively mild conversion of the axonal projection patterns in the knockdown neurons located in L2/3.

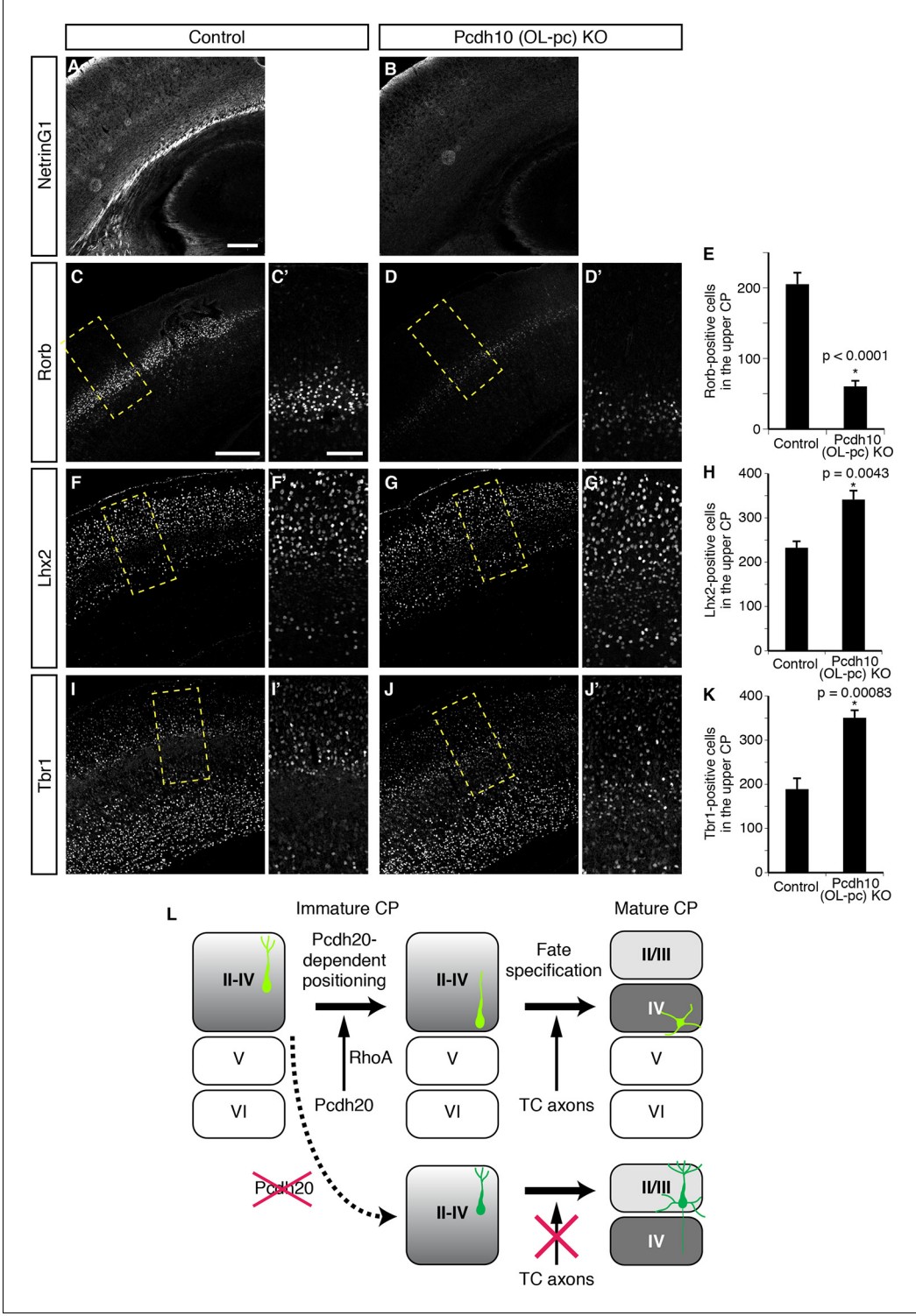

**Figure 7.** Requirement of TCAs for correct differentiation of L4 neurons. (**A–K**) The sections prepared from *Pcdh10* (*OL-pc*) knockout mice were immunostained for NetrinG1 (**A, B**), Rorb (**C, D**), Lhx2 (**F, G**) and Tbr1 (**I, J**). The boxed regions are shown at higher magnification in the columns on the right. Typical images of the P7 cortices were shown. Quantitative analyses were performed on the P3 cortices (**E, H, K**) (n = 6 Control, n = 5 *Pcdh10* (*OL-pc*) KO for Rorb; n = 4 Control, n = 4 *Pcdh10* (*OL-pc*) KO for Lhx2; n = 4 Control, n = 5 *Pcdh10* (*OL-pc*) KO for Tbr1). (**L**) A model for L4 specification suggested by our study is presented. We propose that the identities of the superficial layer neurons are not completely specified before the neurons eventually come to reside beneath the marginal

*Figure 7 continued on next page*

*Figure 7 continued*

zone (left). After radial migration, the future L4 neurons become positioned in the lower part of the superficial cortical plate in a Pcdh20-dependent manner (upper middle). This positioning enables the future L4 neurons to come in contact with TCAs, which appears to be required for further specification and maturation of the L4 neurons (upper right). In the absence of Pcdh20, the neurons reside in the upper part and are unable to receive signals from TCAs (lower row). Scale bars, 200 μm (**A, C**); 100 μm (**C'**).

Our results also revealed the involvement of a mechanism downstream of Pcdh20, namely, RhoA signaling, in the regulation of cell positioning of the future L4 neurons. We observed that the effect of dominant-negative RhoA mimicked the failure in neuronal positioning induced by *Pcdh20* knock-down, and that CA-RhoA was able to restore the phenotypes of the *Pcdh20*-knockdown neurons. The relationship between protocadherin and RhoA has also been shown in other systems (*Chung et al., 2007*; *Unterseher et al., 2004*). Although we do not yet know the mechanism underlying the activation of RhoA by Pcdh20, a mechanism similar to that observed in the regulation of Rho downstream of other protocadherins, such as paraxial protocadherin, may be operative (*Chung et al., 2007*).

How does Pcdh20-RhoA regulate the laminar positioning of the "future L4 neurons"? We assume that the loss of Pcdh20 primarily leads to dysregulation of dendritic morphogenesis, followed by malpositioning of the L4 neurons, which is supported by several lines of evidence. First, *Pcdh20* knockdown in an in vitro culture, where cell positional effects are ignorable, resulted in increased dendritogenesis without altering L4 fate specification. In vivo, Pcdh20- and Dcx double-knockdown neurons tended to exhibit pyramidal morphology even if they were in L4, especially those located near the margin of L4, also suggesting the primary effect, to some extent at least, of Pcdh20 on cell morphology. Regulation of dendritic morphogenesis by another protocadherin was also reported previously (*Shima et al., 2004*). Second, in the time-course experiment of *Pcdh20* knockdown, we noticed that loss of *Pcdh20* first affected the dendrite morphology leading to extensive dendritic arborization, when the control cells already started to exhibit simplification of their complex apical primitive dendrites (*Figure 2J,K*). Finally, active RhoA rescued the malpositioning caused by *Pcdh20* knockdown with reduced arborization of the apical dendrites of the *Pcdh20*-knockdown neurons (*Figure 5C*). Because the arborized dendrites were predominantly found in the MZ, it is possible that the abnormal dendrites might have caused the knockdown neurons to become 'stuck' to the MZ.

How the Pcdh20-RhoA signaling is activated is yet to be defined. Considering that several members of the protocadherin family exhibit homophilic binding activity (*Kim et al., 2007*; *Morishita and Yagi, 2007*; *Suzuki, 2000*; *Takeichi, 2007*), it is conceivable that homophilic interactions may regulate laminar formation through gathering neurons that share the adhesion activity. However, since the interactions through protocadherins, including Pcdh20, are usually very weak or undetectable (K.O. and K.N. unpublished data), there may be other ligands that activate Pcdh20 to regulate neuronal positioning.

Then, what is the mechanism that gives positional cues to immature L4 neurons to exhibit cell-position-dependent subtype specification? Since the acquisition of the L4-specific characteristics (e. g., simplification of apical dendrites) by the "future L4 neurons" coincided with the arrival of the TCAs into the lower part of the CP (*Lopez-Bendito and Molnar, 2003*), we hypothesized that these axons could positively regulate the L4 identity specification. We demonstrated here that TCAs might be involved in correct subtype specification of L4 neurons by the analysis of TCA-deficient *Pcdh10 (OL-pc)* knockout mice. The results of other studies also suggested that defects in TCAs misspecify L4 identity (*Zhou et al., 2010*) and that TCAs instruct the development of modality-specific properties of cortical neurons (*Pouchelon et al., 2014*). Here, we showed that TCA absence resulted in loss of L4 identity and, surprisingly, instead acquisition of some L2/3 characteristics, which recapitulated the phenotypes observed in *Pcdh20* or *Dcx* knockdown, suggesting that TCAs might provide a positional cue to immature L4 neurons. Molecular mechanisms underlying this action still remain to be determined. Although neuronal transmission from TCAs to L4 neurons is an attractive candidate, it may not be required for this action. It was reported that defects in synaptic activity from TCAs to cortical neurons did not appear to be involved in subtype specification of L4 neurons in mice

(*Hannan et al., 2001*; *Iwasato et al., 2000*). A more recent study showed that complete blockade of thalamocortical neurotransmission resulted in perturbed development of L4 neurons. Initial specification of cortical neurons, however, appeared largely normal (*Li et al., 2013*), which occurs through the first week after birth, at the stage when specification failure was already observed in our mouse model.

Finally, another intriguing finding of this study was the apparent involvement of a transmembrane protein in the subtype specification in the post-migratory stage. This suggested that the environment around the immature neurons could also influence the subtype specification of the neurons. Although postmitotic expressions of fate determinants, such as subtype-specific transcription factors, have been observed in the immature CP, it is not known whether these factors are induced intrinsically downstream of the fate determinants in the progenitor cells or in response to the extracellular environment. Therefore, it will be important to examine the role of the environment in the subtype specification of the postmitotic/post-migratory neurons in other contexts.

## Materials and methods

### Animals

Pregnant ICR mice were purchased from Japan SLC (Hamamatsu, Japan). The *Pcdh10 (OL-pc)* knockout mouse line generated by Lexicon Pharmaceuticals (The Woodlands, TX) has been described previously (*Uemura et al., 2007*). The morning of vaginal plug detection was designated as E0.5. All animal experiments were performed under the control of the Keio University Institutional Animal Care and Use Committee in accordance with Institutional Guidelines on Animal Experimentation at Keio University.

### Plasmids

The pSilencer 3.0-H1 plasmid (Ambion, Austin, TX) containing the H1 RNA promoter for the expression of short hairpin RNA (shRNA) was used for construction of the shRNA-encoding plasmids. The inserted sequences were 5'-TATTTCATAGAAGGACTGCACttgatatccgGTGCAGTCCTTCTATGA-AATA-3' (the lower-case letters indicate the linker sequence) for PC20sh, 5'-TATTT**G**ATAG**T**AG-GA**G**TGCACttgatatccgGTGCA**C**TCCT**A**CTAT**C**AAATA-3' (the bold letters indicate mutated nucleotides) for PC20sh_mut, and 5'-TTAGAGCTGCCAGTTATATCCttgatatccgGGATATAACTGGC-AGCTCTAA-3' for PC20UTRsh. The resultant plasmids targeted nt 1620–1640 or nt 3342–3362 of the *Pcdh20* transcript (GenBank Accession No. AK083114).

For expression of GFP or mCherry, the pCAGGS vector (kind gift from J. Miyazaki, Osaka University) carrying the enhanced GFP cDNA (Clontech, Palo Alto, CA) or mCherry (Clontech) was used. For expression of Pcdh20, the gene encoding full-length mouse Pcdh20, obtained from the FAN-TOM RIKEN full-length cDNA clones (AK083114) (*Kawai et al., 2001*; *Okazaki et al., 2002*), was cloned into the plasmid vector, pCAGGS. In the case of expression of Pcdh20 in postmitotic neurons, the pTα1 vector harboring a neuron-specific Tα1 promoter vector was utilized (kind gift from F. D. Miller) (*Gloster et al., 1994*). For expression of hemagglutinin (HA)-tagged Pcdh20, an HA tag was inserted after the signal sequence of Pcdh20. For the rescue experiments, a rescue vector harboring four mutations in the PC20sh targeting site was used. All constructs were verified by nucleotide sequencing. pCAGGS-DN-RhoA and pEF-CA-RhoA were kindly provided by T. Kawauchi (Keio University) and M. Hoshino (National Center of Neurology and Psychiatry), and K. Kaibuchi (Nagoya University), respectively.

Dcx_sh targeting the 3'UTR of the *Dcx* gene and Dcx_sh_m harboring point mutations in Dcx_sh are described elsewhere (*Bai et al., 2003*) (gift from J. LoTurco, University of Connecticut).

### In utero electroporation and BrdU injection

Pregnant mice were deeply anesthetized, and in utero electroporation was carried out as described previously (*Tabata and Nakajima, 2001*). In brief, various shRNA expression vectors (4 mg/ml, unless otherwise mentioned) together with the pCAGGS vector carrying the enhanced GFP cDNA (1 mg/ml) were injected into the lateral ventricle of the intrauterine embryos, and electric pulses (33 V, 50 ms, 4 times) were then applied using an electroporator (CUY-21; NEPA GENE, Chiba, Japan) with a forceps-type electrode (CUY650P3; NEPA GENE).

For injection of BrdU, the pregnant mice were injected intraperitoneally with a BrdU solution (50 mg per kg body weight; Sigma-Aldrich, St. Louis, MO). In the experiments of electroporation and serial injections of BrdU, we judged GFP$^+$/BrdU$^-$ cells as postmitotic cells when electroporation was performed. One concern of this experiment is that few more cell divisions could produce GFP$^+$/BrdU$^-$ cells through dilution of BrdU. However, if the cells divided for a few more cycles, GFP plasmids would simultaneously be diluted. It is thus supposed that the GFP$^+$/BrdU$^-$ cells resulted from dilution of BrdU are, if any, very rare.

## Immunohistochemistry

Embryos and pups were fixed for 5 to 8 hr in phosphate-buffered saline (PBS) containing 4% PFA (wt/vol), incubated overnight at 4°C with 20% sucrose in PBS (wt/vol), embedded in OCT compound (Sakura Finetek, Tokyo, Japan), and sectioned with a cryostat to obtain 14-μm-thick coronal sections. For detection of Rorb and BrdU, antigen retrieval was performed by autoclave treatment of the sections for 5 min at 105°C in 0.01 M sodium citrate buffer (pH 6.0). As primary antibodies, we used mouse antibody to BrdU (BD Biosciences, San Diego, CA), rabbit antibody to GFP (MBL, Nagoya, Japan), chicken antibody to GFP (Abcam, Cambridge, UK), rabbit antibody to Rorb (Diagenode, Leige, Belgium), mouse antibody to Rorb (Perseus Proteomics, Tokyo, Japan), goat antibody to Lhx2 (Santa Cruz, Santa Cruz, CA), mouse antibody to Satb2 (Abcam), goat antibody to KCNH5 (Santa Cruz), goat antibody to NetrinG1 (R&D systems, Minneapolis, MN) and rabbit antibody to Tbr1 (kind gift from R. Hevner, University of Washington). Immune complexes were detected with FITC–, TRITC– or Cy5–conjugated secondary antibodies (Jackson ImmunoResearch Laboratories, West Grove, PA). For nuclear staining, we used 1 μg/ml propidium iodide (PI; Molecular Probes, Eugene, OR) and 2 μg/ml DAPI (Molecular Probes). Images were acquired using confocal microscopes (FV300 and FV1000; Olympus, Tokyo, Japan).

## In situ hybridization

In situ hybridization of frozen sections was performed as described previously (*Tachikawa et al., 2008*). Probes for Pcdh20 were prepared from the FANTOM clone set (*Kawai et al., 2001*; *Okazaki et al., 2002*).

## Quantitative RT-PCR analysis

Total RNA was extracted from the neocortical tissues at various developmental stages using Trizol reagent (Invitrogen, Carlsbad, CA), followed by DNase treatment, and subjected to RT with an oligo (dT)$_{12-18}$ primer (Invitrogen). The resulting cDNA was subjected to real-time PCR in ABI PRISM 7500 (Applied Biosystems, Foster City, CA) using the Universal ProbeLibrary system (Roche, Basel, Switzerland). The expression level of *Pcdh20* mRNA was normalized relative to that of *β-actin* mRNA. The sense primer, antisense primer and, Universal ProbeLibrary number, respectively, were as follows: *Pcdh20*, 5'-CTGAAGAGTGCGATGTTTCG-3', 5'-AGTGCAGGAGGAAAGCAAAC-3' and probe #15; *β-actin*, 5'-CTAAGGCCAACCGTGAAAAG-3', 5'-ACCAGAGGCATACAGGGACA-3' and probe #64.

## Transfection and immunoblotting

293T cells were grown in Dulbecco's modified Eagle's medium (DMEM) containing penicillin-streptomycin and 10% fetal bovine serum (vol/vol). The cells were transfected with various plasmids using GeneJuice (Novagen, Darmstadt, Germany) for 24 hr, and lysed as described previously (*Oishi et al., 2009*).

For transfection into primary cortical cells, E15.5 cortices were dissociated with trypsin, and plasmids were introduced into the dissociated cells using the Amaxa Nucleofector system (Lonza, Basel, Switzerland). The cells were cultured in Neurobasal medium containing 2% B27 and 0.4 mM L-glutamine. The lysates were then subjected to immunoblot analysis. The blots were probed with antibodies to HA (Y11, Santa Cruz), Pcdh20 (Abnova, Taipei, Taiwan or Abgent, San Diego, CA) and GFP (MBL).

### Fluoro-Gold injection

Pups were anesthetized, and Fluoro-Gold (40 mg/ml; Fluorochrome, LLC) was injected into the neocortex contralateral to the side of electroporation. Three or four injections per pup were administered, using about 200 nl of Fluoro-Gold solution per injection site. Three days later, the pups were fixed and analyzed.

### Quantitative analysis of the brain slices

To quantify the pattern of migration, the position of each GFP-positive cell relative to the total distance from the bottom of the CP decided by the subplate to the outer edge of the CP (pial surface) was measured using the Image J software (National Institutes of Health shareware program), followed by sorting into 10 bins. To quantify cell differentiation and Fluoro-Gold assays, the number of total GFP-positive cells and that of a given maker- or Fluoro-Gold-positive cells among GFP-positive cells were counted, and the percentages were calculated. For the analysis of *Pcdh10 (OL-pc)* knockout mice, the upper CP was first decided from DAPI staining, and the number of a given marker-positive cells was counted in 600-μm width of cortical columns. For all assays, more than 200 cells were counted for each sample. We repeated all experiments using at least two different litters, and at least four sections from at least three independent brains were analyzed for quantification. We analyzed the somatosensory area of the neocortex.

### Electrophysiology

Coronal slices of the somatosensory cortex (300 μm thick) were prepared from P14-19 mice under deep anesthesia with isoflurane, and placed in artificial cerebrospinal fluid (ACSF) containing (in mM): 126 NaCl, 3 KCl, 1.3 $MgSO_4$, 2.4 $CaCl_2$, 1.2 $NaH_2PO_4$, 26 $NaHCO_3$, and 10 glucose, as described previously (*Yoshimura et al., 2003*). For whole-cell recordings, GFP-positive neurons were detected by fluorescence observations and recorded under infrared differential interference contrast (IR-DIC) optics (BX51WI, Olympus), with patch pipettes (4–6 MΩ) filled with a solution containing (mM) 130 K-gluconate, 8 KCl, 1 $MgCl_2$, 0.6 EGTA, 10 HEPES, 10 Na-phosphocreatine, 3 MgATP and 0.5 $Na_2GTP$ (pH 7.4 with KOH). Biocytin (0.2%) was included in the solution to label the recorded cells intracellularly. We selected cells with a high seal resistance (>1 GΩ) and a series resistance (<30 MΩ) for the analysis. Membrane properties were assessed by injecting currents in current-clamp recordings or by applying step pulses in voltage-clamp recordings. Spontaneous EPSCs were recorded at -65 mV. After the recordings, the slices were fixed and resectioned. The sections were processed by a method using Cy3-conjugated streptavidin, and labeled neurons were imaged with a Zeiss LSM510 (Oberkochen, Germany) confocal microscope. The number of recorded cells in each condition was described in the figure.

### In vitro culture

Primary neocortical neuronal cultures were prepared as described (*Oishi et al., 2009*). Briefly, E18.0 neocortices that had been electroporated with shRNA vectors on E14.0 were dissociated with 0.125% trypsin and plated on to poly-D-lysine-coated dishes. The cells were cultured for the indicated period, fixed, and subjected to immunocytochemistry for GFP (MBL) or Rorb (Perseus Proteomics).

### Statistical analysis

Data were represented as means ± SEM. Statistical analyses were performed using the two-tailed Welch's t-test. Differences between groups were considered to be significant at p<0.05. Each p value was stated in figures or figure legends.

## Acknowledgements

This work was supported by the Strategic Research Program for Brain Sciences ("Understanding of molecular and environmental bases for brain health"), the Grants-in-Aid for Scientific Research of the Ministry of Education, Culture, Sports, Science, and Technology of Japan (15H02355, 22111004, 25640039, 15H01586, 19700295, 21700356), Takeda Science Foundation, Naito Foundation, Terumo Life Science Foundation, Life Science Foundation of Japan, Keio Gijuku Fukuzawa Memorial Fund for

the Advancement of Education and Research, and Keio Gijuku Academic Development Funds. We thank T Shimogori for critical reading and valuable comments, J Miyazaki, J LoTurco, FD Miller, T Kawauchi, M Hoshino, K Kaibuchi, and R Hevner for the reagents, as well as members of the Nakajima laboratory for the discussions and technical assistance.

## Additional information

### Funding

| Funder | Grant reference number | Author |
|---|---|---|
| Ministry of Education, Culture, Sports, Science, and Technology | 15H02355, 22111004, 25640039, 15H01586, 19700295, 21700356 | Koji Oishi Kazunori Nakajima |
| Takeda Science Foundation | | Kazunori Nakajima |
| Strategic Research Program for Brain Sciences | Understanding of molecular and environmental bases for brain health | Kazunori Nakajima |
| Terumo Life Science Foundation | | Kazunori Nakajima |
| Life Science Foundation of Japan | | Kazunori Nakajima |
| Keio University | Keio Gijuku Fukuzawa Memorial Fund for the Advancement of Education and Research | Kazunori Nakajima |
| Keio University | Keio Gijuku Academic Development Funds | Koji Oishi Kazunori Nakajima |
| Naito Foundation | | Kazunori Nakajima |

The funders had no role in study design, data collection and interpretation, or the decision to submit the work for publication.

### Author contributions

KO, Conception and design, Acquisition of data, Analysis and interpretation of data, Drafting or revising the article, Contributed unpublished essential data or reagents; NN, Acquisition of data, Analysis and interpretation of data, Drafting or revising the article; KT, SS, Acquisition of data, Drafting or revising the article, Contributed unpublished essential data or reagents; MA, Acquisition of data, Drafting or revising the article; SH, NY, Analysis and interpretation of data, Drafting or revising the article, Contributed unpublished essential data or reagents; YY, Analysis and interpretation of data, Drafting or revising the article; KN, Conception and design, Analysis and interpretation of data, Drafting or revising the article

### Author ORCIDs

Kazunori Nakajima, http://orcid.org/0000-0003-1864-9425

### Ethics

Animal experimentation: All animal experiments were performed under the control of the Keio University Institutional Animal Care and Use Committee in accordance with Institutional Guidelines on Animal Experimentation at Keio University (approved protocol number 08065-(9)).

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
