## [Decision Letter]

Thank you for submitting your work entitled "Identity of neocortical layer 4 neurons is specified through correct positioning into the cortex" for consideration by *eLife*. Your article has been reviewed by three peer reviewers, and the evaluation has been overseen by Joseph G Gleeson (Reviewing Editor) and K VijayRaghavan as the Senior Editor.

The reviewers have discussed the reviews with one another and the Reviewing editor has drafted this decision to help you prepare a revised submission. While we are favorably inclined there are some significant shortcomings in the study that need to be fully addressed for acceptance. The goal of *eLife* is to avoid requesting revisions that would take more than 2 months, and the reviewers estimated that these suggestions should be possible to address within this time frame.

Summary:

This is an interesting study that seeks to alter the longstanding paradigm used for understanding cortical neuron fate. Authors provide evidence to show that *Pcdh20* has a role in regulating the ultimate fate of L4 neurons and that this action is a) post-mitotic and b) acts via controlling laminar positioning. The authors provide a spectrum of evidence to support these ideas, however, all reviewers believe that there are some significant shortcomings in the study.

Essential revisions:

The main criticism focuses on three points that should be addressed in a revised manuscript.

The reliance on shRNA has left reviewers with doubt that the protein is inactivated. The authors rely very heavily on shRNA knockdown of *Pcdh20*. They show that they can knock this down in vitro but the effect is far from complete. The authors show that the cells still express 20% residual protein. This is in the optimal conditions of transfection of cells in cell culture with tagged *Pcdh20*. The authors never show the degree of protein knockdown in the cortex when they use this approach. They simply must develop a way to show that their key manipulation (on which they base the majority of their results) actually has the intended effect of decreasing expression in the relevant cells. Without this demonstration it is hard to really know what is going on. The reviewers would like to see experiments that show that protein levels of PCDH20 are indeed reduced. One possible independent way, which the authors may like to consider, is the following: Late in the study the authors use a mouse with genetic disruption of the *Pcdh20* locus. We realize that these mice are hard to interpret due to the disruption of TCA fibers but the authors could perhaps consider using a cell transplantation approach using cells from these mutants. This would further establish the role for *Pcdh20* rather than shRNA manipulation in the phenotype.

We have important concerns about the Rho experiments, which require tempered interpretation, because the use of DN and CA constructs is likely to lead to many migratory effects that are distinct from the *Pcdh20* loss-of-function effects. These molecules are downstream of many proteins that act at various stages in the migrating neurons. This makes these experiments difficult to interpret. While one possibility, only slightly better, is that the authors could transfect with T1-α promoter driving these constricts. Consistent with these likely global effects, these manipulations lead to somewhat different abnormal cellular distributions (according to the authors; own data). For example, in Figure 5 the electroporated cells with DN-RhoA are more widely distributed, including into deeper layers. This implies a migration phenotype from this manipulation. This could be a major problem with this portion of the study and needs to be carefully addressed by altering the interpretation so that these concerns are well reflected and conclusions tempered.

PCDH20 could function directly on fate or indirectly on fate through an effect on positioning. We suggest testing the effects of PCDH20 on neurons ex vivo (in culture). The authors looked at dendritic patterns (which are altered), but not at molecular fate marker patterns: they should do this, as if PCDH20 does not alter molecular fate markers in ex vivo experiments, it would strongly suggest that neuronal positioning is the crucial event, while if the fate is somehow altered (more expression of layer 4 markers) it would be more compatible with a direct effect on PCDH20. This could be combined with BrdU birthdating to label only cohorts of neurons to maximize specificity and sensitivity.

---

## [Author Response]

*The reliance on shRNA has left reviewers with doubt that the protein is inactivated. The authors rely very heavily on shRNA knockdown of Pcdh20. They show that they can knock this down* in vitro

*but the effect is far from complete. The authors show that the cells still express 20% residual protein. This is in the optimal conditions of transfection of cells in cell culture with tagged Pcdh20. The authors never show the degree of protein knockdown in the cortex when they use this approach. They simply must develop a way to show that their key manipulation (on which they base the majority of their results) actually has the intended effect of decreasing expression in the relevant cells. Without this demonstration it is hard to really know what is going on. The reviewers would like to see experiments that show that protein levels of PCDH20 are indeed reduced. One possible independent way, which the authors may like to consider, is the following: Late in the study the authors use a mouse with genetic disruption of the Pcdh20 locus. We realize that these mice are hard to interpret due to the disruption of TCA fibers but the authors could perhaps consider using a cell transplantation approach using cells from these mutants. This would further establish the role for Pcdh20 rather than shRNA manipulation in the phenotype.*

We really agree about this point. We now have performed Western blot analysis using primary cortical cultures to see if the endogenous protein levels of *Pcdh20* are affected by knockdown. The result showed that the *Pcdh20* protein level is indeed reduced in the *Pcdh20* knockdown cells (presented in new Figure 2). As you may notice, there are still remaining *Pcdh20* proteins in the knockdown cells probably due to imperfect transfection efficiency in this experiment (roughly 80% transfection efficiency) as well as incomplete knockdown effects.

We also conducted another independent shRNA experiment using another shRNA targeting *Pcdh20*. The result showed that this shRNA phenocopied the original shRNA phenotype (new Figure 2).

We believe that these new results as well as the original ones strongly support the notion that the phenotypes observed here is actually by the reduction of *Pcdh20* protein levels but not due to off-target effects.

*We have important concerns about the Rho experiments, which require tempered interpretation, because the use of DN and CA constructs is likely to lead to many migratory effects that are distinct from the Pcdh20 loss-of-function effects. These molecules are downstream of many proteins that act at various stages in the migrating neurons. This makes these experiments difficult to interpret. While one possibility, only slightly better, is that the authors could transfect with T1-*α *promoter driving these constricts. Consistent with these likely global effects, these manipulations lead to somewhat different abnormal cellular distributions (according to the authors; own data). For example, in*
Figure 5
*the electroporated cells with DN-RhoA are more widely distributed, including into deeper layers. This implies a migration phenotype from this manipulation. This could be a major problem with this portion of the study and needs to be carefully addressed by altering the interpretation so that these concerns are well reflected and conclusions tempered.*

We constructed several expression vectors based on the Cre-loxP system, in which DN-RhoA is expressed under the CAG promoter only after a loxP-stop-loxP cassette is excised by the Cre recombinase, which is driven by Talpha1 promoter in this case. However, in any of these cases, the levels of RhoA expression were found to be very low compared to the original pCAGGS-DN-RhoA construct for some reason. So, unfortunately we could not address the issue using this system.

Instead, we examined the effect of DN-RhoA at E18.0, when future L4 neurons just finished migration and started to mature. We found that there was no significant difference in cell positioning between control and DN-RhoA expressing cells at least at the concentration/condition we used (new Figure 5—figure supplement 1).

The role of RhoA signaling in neuronal migration is controversial; inhibition of RhoA interfered with neuronal migration in some studies (Pacary et al., 2011, PMID:21435554; Xu et al., 2015, PMID: 26511243), but it promoted migration in other cases (Cappello et al., 2012, PMID:22405202; Nguyen et al., 2006, PMID:16705040; Ge et al., 2006, PMID:16432194). These previous discrepancies might have been caused by the differences in their experimental conditions. From our new analysis, the DN-RhoA expressing neurons appeared to settle in the ridge of the CP very similarly to normal cells after radial migration, although they could have migrated differently. Therefore, we think that the phenotype observed at P7 (Figure 5) is most probably due to the event after radial migration is completed by around E18.0 rather than mainly caused by the migration difference.

However, we do not exclude the possibility that there might be an additional effect of migration difference on subtype specification, especially in the case of enhanced migration, because there tended to be more cells in deeper layers with DN-RhoA at P7 (Figure 5; this is not statistically significant though). One might suspect that accelerated neuronal migration could cause premature differentiation, which would result in more cells in deeper layers at P7. This effect of DN-RhoA, if any, was thought to be minor, though. It would be interesting to address this issue in future studies by looking at subtype specification of the cells that migrate much faster than normal cells by some treatment. We have added this point in the text (subsection “Involvement of RhoA signaling in the *Pcdh20*-dependent positioning of L4 neurons”, third paragraph). We highly appreciate this important comment by the reviewers/editor.

*PCDH20 could function directly on fate or indirectly on fate through an effect on positioning. We suggest testing the effects of PCDH20 on neurons ex vivo (in culture). The authors looked at dendritic patterns (which are altered), but not at molecular fate marker patterns: they should do this, as if PCDH20 does not alter molecular fate markers in ex vivo experiments, it would strongly suggest that neuronal positioning is the crucial event, while if the fate is somehow altered (more expression of layer 4 markers) it would be more compatible with a direct effect on PCDH20. This could be combined with BrdU birthdating to label only cohorts of neurons to maximize specificity and sensitivity.*

This suggestion is very helpful to further clarify whether *Pcdh20* indeed functions indirectly for subtype specification. We cultured future L4 neurons (E14.0-born) from E18.0 cortex, when they are not yet matured, and looked at the subtype specification after 4 days in vitro, when these cells become *Rorb*-positive in vivo. To label future L4 neurons and introduce knockdown vectors, electroporation with control or *Pcdh20*-knockdown vector was performed at E14.0, and these cells were utilized for further in vitro cultures. We did not find any difference of *Rorb* expression between control and *Pcdh20*-knockdown neurons in this condition (new Figure 6; subsection “*Pcdh20* specifies L4 identity through controlling cell positioning“, last paragraph). Together with the result shown in Figure 6 that *Pcdh20*-knockdown neurons in a similar condition had more arborized dendrites, this result further supports the notion that *Pcdh20* directly regulates cell morphology rather than cell fate. This discussion was also added in the text (Discussion, second and sixth paragraphs).